# The Anti-*Leishmania amazonensis* and Anti-*Leishmania chagasi* Action of Copper(II) and Silver(I) 1,10-Phenanthroline-5,6-dione Coordination Compounds

**DOI:** 10.3390/pathogens12010070

**Published:** 2023-01-01

**Authors:** Simone S. C. Oliveira, Vanessa S. Santos, Michael Devereux, Malachy McCann, André L. S. Santos, Marta H. Branquinha

**Affiliations:** 1Laboratório de Estudos Avançados de Microrganismos Emergentes e Resistentes (LEAMER), Departamento de Microbiologia Geral, Instituto de Microbiologia Paulo de Góes (IMPG), Universidade Federal do Rio de Janeiro (UFRJ), Rio de Janeiro 21941-901, Brazil; 2The Inorganic Pharmaceutical and Biomimetic Research Centre, Focas Research Institute, Dublin Institute of Technology, D08 CKP1 Dublin, Ireland; 3Chemistry Department, Maynooth University, National University of Ireland, W23 F2H6 Maynooth, Ireland; 4Programa de Pós-Graduação em Bioquímica, Instituto de Química, Universidade Federal do Rio de Janeiro (UFRJ), Rio de Janeiro 21941-909, Brazil

**Keywords:** leishmaniasis, coordination compounds, 1,10-phenanthroline-5,6-dione, metal ions, proliferation, apoptosis

## Abstract

Leishmaniasis is a neglected disease caused by protozoa belonging to the *Leishmania* genus. Notably, the search for new, promising and potent anti-*Leishmania* compounds remains a major goal due to the inefficacy of the available drugs used nowadays. In the present work, we evaluated the effects of 1,10-phenanthroline-5,6-dione (phendione) coordinated to silver(I), [Ag(phendione)_2_]ClO_4_ (Ag-phendione), and copper(II), [Cu(phendione)_3_](ClO_4_)_2_·4H_2_O (Cu-phendione), as potential drugs to be used in the chemotherapy against *Leishmania amazonensis* and *Leishmania chagasi*. The results showed that promastigotes treated with Ag-phendione and Cu-phendione presented a significant reduction in the proliferation rate. The IC_50_ values calculated to Ag-phendione and Cu-phendione, respectively, were 7.8 nM and 7.5 nM for *L. amazonensis* and 24.5 nM and 20.0 nM for *L. chagasi*. Microscopical analyses revealed several relevant morphological changes in promastigotes, such as a rounding of the cell body and a shortening/loss of the single flagellum. Moreover, the treatment promoted alterations in the unique mitochondrion of these parasites, inducing significant reductions on both metabolic activity and membrane potential parameters. All these cellular perturbations induced the triggering of apoptosis-like death in these parasites, as judged by the (i) increased percentage of annexin-positive/propidium iodide negative cells, (ii) augmentation in the proportion of parasites in the sub-G_0_/G_1_ phase and (iii) DNA fragmentation. Finally, the test compounds showed potent effects against intracellular amastigotes; contrarily, these molecules were well tolerated by THP-1 macrophages, which resulted in excellent selective index values. Overall, the results highlight new selective and effective drugs against *Leishmania* species, which are important etiological agents of both cutaneous (*L. amazonensis*) and visceral (*L. chagasi*) leishmaniasis in a global perspective.

## 1. Introduction

Leishmaniasis is a tropical neglected disease caused by more than 20 species of protozoan parasites of the *Leishmania* genus that are transmitted by the bite of more than 90 species of sand flies from the *Phlebotomus* genus. Besides its endemic profile in 98 countries, leishmaniasis represents a serious public health concern in terms of control of transmission reservoirs [1]. Leishmaniasis presents a wide spectrum of clinical manifestations that appear to result from a combination of intrinsic properties of the parasite strain and species as well as various factors of the vertebrate host, such as age, genetic predisposition and immunological status [1,2]. In this regard, this disease can be traditionally classified into three main clinical forms: visceral, cutaneous (localized or diffuse) and mucocutaneous, which differ in their immunopathology, degree of morbidity and mortality [1,2].

Different chemotherapeutics with distinct mechanisms of action are available for the current treatment of leishmaniasis. Pentavalent antimonials, such as *N*-methyl glucamine antimoniate (glucantime^®^) and sodium stibogluconate (pentostan^®^) are the first-choice drugs. Amphotericin B, miltefosine and paromomycin are used as second options for treatment [1,3,4]. However, all antileishmanial therapies are very problematic, mainly due to the extensive toxicity, lack of efficacy, the necessity of parenteral administration, high costs, the emergence of drug resistance and a lack of access in regional areas [3,4]. Antimonials, which have been used in the treatment of leishmaniasis since the beginning of the last century, are responsible for important side effects in the host, and the resistance of the parasite to these drugs has been reported in several areas of the world. Miltefosine is an oral drug with few side effects, but it is extremely expensive, as are the lipid formulations of amphotericin B, which have many side effects and resistant parasites are usually reported [3,4].

In this scenario, coordination compounds have played an important role in the development of new chemotherapeutics with pharmacological applications [5,6]. Coordination compounds’ synthesis is relatively easy and the geometric possibilities resulting from the use of a metal center make this an attractive approach for the development of novel pharmacological agents [6]. The chemistry of metal complexes with heterocyclic linkers has attracted considerable interest in recent years, becoming a growing class of research due to the prospect of synthesis of a large number and a wide variety of synthetic binders (chelates), which behave like coordination agents for metal ions [6]. 

One of the oldest and best-studied *N*-heterocyclic chelating agents is 1,10-phenanthroline (phen), which acts as a platform for the synthesis of new reagents for biotechnological and medical purposes. Several published works have shown that phen-based compounds have antimicrobial properties against bacteria, fungi and protozoa, including different species of *Leishmania* [6,7,8]. 1,10-Phenanthroline-5,6-dione (phendione) was synthesized through the addition of two carbonyl groups attached at the 5,6-positions. Previous studies on the antibacterial properties of metal-free compounds, including heteroaromatic derivatives of phenanthrene, such as phen and phendione, have revealed that the inclusion of *N*-atoms in the phenanthrene ring has considerably increased its antimicrobial activity. In the search of new bioactive compounds derived from phen or phendione, a previous study described the synthesis of a series of new compounds, including [Cu(phendione)_3_](ClO_4_)_2_·4H_2_O(Cu-phendione) and [Ag(phendione)_2_]ClO_4_ (Ag-phendione), presenting a greater efficacy and a lower toxicity to multicellular organisms [9,10]. In this context, in vivo tests showed that Cu-phendione and Ag-phendione had a low toxicity for Swiss mice and *Galleria mellonella* larvae [6]. In addition, phendione-derived drugs, in both complexed or metal-free forms, were able to alter the functioning of a variety of microbial systems [6,8,11,12,13], such as bacteria (*Acinetobacter baumannii* [13], *Escherichia coli* [14], *Klebsiella pneumoniae* [15] and *Pseudomonas aeruginosa* [9]), yeasts (*Candida albicans* [6] and non-*albicans Candida* species [16]), filamentous fungi (*Pseudallescheria boydii* [6] and *Phialophora verrucosa* [11]) and parasites (*Trichomonas vaginalis* [12] and *Leishmania braziliensis* [17]). 

Recently, our group showed that both Cu-phendione and Ag-phendione displayed a leishmanicidal effect against promastigote forms of *L. braziliensis* [17]. In this context, the present work aimed to amplify the studies with these potent compounds against other *Leishmania* species, since the extreme heterogeneous profile of leishmaniasis culminates in distinct responses to drugs. Based on this premise, *L. amazonensis* and *L. chagasi*, relevant etiologic agents of cutaneous and visceral forms of leishmaniasis, respectively, were selected to evaluate their susceptibility to the treatment with phendione and its silver and copper derivatives as well as to decipher the mechanisms of action and to elucidate the triggering death pathway. In addition, the effects of these test compounds on *Leishmania*-macrophage interaction were evaluated.

## 2. Materials and Methods

### 2.1. Parasites and Cultivation

*Leishmania amazonensis* (MHOM/BR/PH8) and *Leishmania chagasi* (MHOM/BR/1974/PP75) were obtained from Coleção de *Leishmania* from Fundação Oswaldo Cruz (FIOCRUZ; *Leishmania* Type Culture Collection-LTTC-WDCM 731), Rio de Janeiro, Brazil. Promastigotes were grown in Schneider’s insect medium (Sigma-Aldrich, St Louis, MO, USA), pH 7.2, containing 10% heat-inactivated fetal bovine serum (FBS) (Cultilab, São Paulo, SP, Brazil) at 28 °C.

### 2.2. Macrophages Cultivation

Human leukemia monocytic cell line (THP-1) was maintained in 25 cm^2^ tissue culture flasks with RPMI 1640 medium (Sigma-Aldrich, St Louis, MO, USA) supplemented with 10% FBS at 37 °C in an atmosphere containing 5% CO_2_. The culture medium was exchanged every three days. For interaction experiments, THP-1 cells in 24- or 96-well plates (2 × 10^5^ cells/well) were differentiated into macrophages by treatment with phorbol-12-myristate-13-acetate (PMA; 40 ng/mL) (Sigma-Aldrich, St Louis, MO, USA) for 48 h. Then, the plates were washed twice with sterile phosphate-buffered saline (PBS; pH 7.2) to remove PMA and a new RPMI 1640 medium was added [18]. Differentiated cells, used in all experiments, showed similar morphological changes and the ability to adhere to the culture plates as macrophages.

### 2.3. Test Compounds

1,10-Phenanthroline-5,6-dione (phendione), [Cu(phendione)_3_](ClO_4_)_2_·4H_2_O (Cu-phendione) and [Ag(phendione)_2_]ClO_4_ (Ag-phendione) were prepared in accordance to published methods [19]. The simple salts AgNO_3_ and CuSO_4_ as well as 1,10-phenanthroline (phen) (Sigma-Aldrich, St Louis, MO, USA) were used as appropriated controls.

### 2.4. Effects of Coordination Compounds on Promastigotes’ Growth Rate

Promastigotes were counted using a Neubauer chamber and resuspended in fresh medium at a final concentration of 5 × 10^5^ viable promastigotes per milliliter. The viability was assessed by mobility and a lack of staining after challenging with Trypan blue. Each metal-coordinated compound was added to the cultures at final concentrations ranging from 2.5 to 30 nM for *L. amazonensis* and from 5 to 50 nM for *L. chagasi*, starting from a stock solution in dimethyl sulfoxide (DMSO; Sigma-Aldrich, St Louis, MO, USA). Phendione and phen were added to the cultures at final concentrations ranging from 5 to 50 nM and from 400 to 2000 nM, respectively, for both *Leishmania* species. After 72 h of incubation at 28 °C, the number of viable parasites was estimated. The 50% inhibitory concentration (IC_50_), i.e., the drug concentration that caused a 50% reduction in survival/viability, was determined by a linear regression analysis, by plotting the log number of promastigotes versus drug concentration using GraphPad Prism 5 computer software.

### 2.5. Protocol of Parasite Treatment: Looking for Potential Mechanisms of Action

For all the subsequent experiments (except when discriminated), promastigotes (5 × 10^5^ cells/mL) were treated (or not) with each coordination compound for 72 h at concentrations corresponding to ½ × IC_50_, IC_50_ or 2 × IC_50_ values in order to detect the influence of these test compounds on the (i) morphology/morphometry, (ii) ultrastructure, (iii) general metabolism, (iv) mitochondrial dehydrogenase activity, (v) mitochondrial membrane potential, (vi) phosphatidylserine externalization, (vii) incorporation of propidium iodide, (viii) cell cycle, (ix) DNA fragmentation and (x) interaction with macrophages.

### 2.6. Morphology, Morphometry and Ultrastructure

Morphological alterations were evidenced by flow cytometry (FACSCalibur, BD Bioscience, Franklin Lakes, NJ, USA) using a two-parameter histogram of forward scatter (FSC) versus side scatter (SSC) to measure two morphological parameters, cell size and granularity, respectively. An ultrastructural analysis was performed by scanning electron microscopy (SEM). To do it, parasites were fixed for 40 min at 25 °C with 2.5% glutaraldehyde in 0.1 M cacodylate buffer, pH 7.2. After fixation, the parasites were washed in a cacodylate buffer and postfixed with a solution of 1% OsO_4_, 0.8% potassium ferrocyanide and 5 mM CaCl_2_ in the same buffer for 20 min at 25 °C. The parasites were dehydrated in graded series of acetone (30–100%) and then dried by the critical point method, mounted on stubs, coated with gold (20–30 nm) and observed in a Jeol JSM 6490LV scanning electron microscope (JEOL Inc., Peabody, MA, USA). 

### 2.7. General Metabolism

The general metabolism of the parasites was evaluated by a resazurin dye/Alamar blue (7-hydroxy-3H-phenoxazin-3-one-10-oxide) assay (Sigma-Aldrich, St Louis, MO, USA) [20]. *L. amazonensis* and *L. chagasi* promastigotes were incubated in sterile 96-well plates (in a total volume of 100 μL culture medium/well) and then resazurin was added to a final concentration of 0.0125% in PBS [21]. After a 4 h incubation at room temperature, parasites were analyzed in a microplate reader (SpectraMax spectrofluorometer, Molecular Devices, San Jose, CA, USA) using a pair of 590 and 544 nm as emission and excitation wavelengths, respectively. The viability was evaluated based on a comparison with untreated control cells. Parasites were also treated with sodium azide (40 µM) for 30 min in order to obtain nonviable cells to use as a positive control in the viability test.

### 2.8. Mitochondrial Dehydrogenases

The mitochondrial dehydrogenase activity was measured by using 3-(4,5-dimethylthiazol-2-yl)-2,5-diphenyltetrazolium bromide (MTT) (Sigma-Aldrich, St Louis, MO, USA). This reagent was added to a final concentration of 0.5 mg/mL in the culture medium and the parasites were incubated for 4 h in the dark at 28 °C. Then, the MTT solution was removed and 200 µL of DMSO was added to solubilize the formazan crystals. The mitochondrial metabolism was determined spectrophotometrically at 490 nm (SpectraMax Gemini 190, Molecular Devices, San Jose, CA, USA). Parasite cells incubated with sodium azide at 40 µM for 30 min were used as a positive control. 

### 2.9. Mitochondrial Membrane Potential

In order to analyze the mitochondrial membrane potential (ΔΨm), parasites were incubated with 10 µg/mL Rhodamine 123 (R123) (Sigma-Aldrich, St Louis, MO, USA) for 30 min and washed three times with PBS, resuspended in the same buffer and analyzed by flow cytometry (FACSCalibur, BD Bioscience, Franklin Lakes, NJ, USA) equipped with a 15 mW argon laser emitting at 488 nm. As positive control of the depolarization of the mitochondrial membrane, parasites were incubated for 30 min with 2 μM carbonyl cyanide 4-(trifluoromethoxy) phenylhydrazone (FCCP) (Sigma-Aldrich, St Louis, MO, USA), a mitochondrial protonophore. Data represented the analysis of 10,000 events and the results were expressed as the mean of the fluorescence intensity (MFI).

### 2.10. Phosphatidylserine Externalization and Incorporation of Propidium Iodide

Phosphatidylserine externalization and the passive incorporation of propidium iodide (PI) were evidenced by using the double labeling with annexin V conjugated to Alexa-Fluor and PI (Thermo Fisher Scientific, Waltham, MA, SUA). Parasite cells were washed twice with PBS and resuspended in an annexin-V binding buffer (10 mM HEPES, 140 mM NaCl, 2.5 mM CaCl_2_, pH 7.4). Annexin V conjugated to Alexa-Fluor was added and the promastigotes were incubated for 20 min in the dark. After, parasite cells were washed with PBS and incubated with PI for 10 min. Finally, parasite cells were washed, resuspended in PBS and the intensity of the labeling of annexin V and PI was recorded in a FACSCalibur (BD Bioscience, Franklin Lakes, NJ, USA) flow cytometer and analyzed with Flowing software. Data represented the analysis of 10,000 events and the results were expressed as percentage of fluorescent cells (% FC).

### 2.11. Cell Cycle

To analyze the parasite cell cycle, promastigotes were fixed in 70% chilled methanol (diluted in PBS) overnight at −20 °C. Fixed promastigotes were washed thoroughly and then resuspended in 0.5 mL of PI (10 µg/mL in PBS) containing RNase A (100 µg/mL) (Thermo Fisher Scientific, Waltham, MA, SUA), and the mixture was incubated for 20 min in the dark at room temperature. The fluorescence intensity of PI was analyzed with a FACSCalibur (BD Bioscience, Franklin Lakes, NJ, USA) flow cytometer and CellQuest software. Data represented the analysis of 10,000 events and the results were expressed as % FC.

### 2.12. DNA Fragmentation

The detection of DNA fragmentation was performed by using a Terminal Deoxynucleotidyltranferase (TdT)-mediated dUTP Nick End Labeling (TUNEL) assay (Invitrogen, Carlsbad, CA, USA) following the manufacturer’s protocol. Briefly, parasites were incubated in ice-cold 70% ethanol overnight at −20 °C. Then, the parasites were washed twice with PBS for ethanol removal and incubated with a DNA-labeling solution (TdT+ Brd UTP) for 60 min at 37 °C in a water bath. Finally, the parasites were resuspended in PBS and analyzed using the FACSCalibur (BD Bioscience, Franklin Lakes, NJ, USA) flow cytometer using Flowing software. Data represented the analysis of 10,000 events and the results were expressed as % FC.

### 2.13. Macrophage Toxicity

The effects of Cu-phendione and Ag-phendione on the viability of THP-1 cells were evaluated by MTT assay. THP-1 cells (2 × 10^5^ cells/mL) were differentiated, as described previously, in 96-well culture plates in RPMI 1640 medium supplemented with 10% FBS. Then, each metal-coordinated compound was added to the cultures at final concentrations ranging from 0.1 to 3 µM and the macrophage cells were incubated for 24 h at 37 °C in a 5% CO_2_ atmosphere. Afterwards, an MTT reduction assay was carried out as described in Section 2.8. The 50% cytotoxic concentration (CC_50_) was determined by a linear regression analysis after 24 h of treatment with each test compound. 

### 2.14. Leishmania-Macrophage Interaction

THP-1 cells (5 × 10^5^ cells) were differentiated, as described in Section 2.2, in 24-well culture plates containing sterile cover glasses in each well. Then, *L. amazonensis* or *L. chagasi* stationary-phase promastigotes, obtained after 3 days of culture, were used to infect the mammalian cells (10 parasites per macrophage) for 24 h at 37 °C, 5% CO_2_, to allow parasite internalization. Afterwards, the cultures were washed with sterile PBS to remove noninternalized parasites and then fresh medium with 10% FBS was added. Infected THP-1 cells were treated for 24 h, under the same conditions, with concentrations that maintained 95% viability of the host cell: 0.025, 0.05 and 0.1 µM of Ag-phendione and 0.04, 0.08 and 0.16 µM of Cu-phendione. After this period, the cultures were fixed with methanol and stained with Giemsa. The percentage of infected macrophages was determined by randomly counting 200 cells on each of triplicate coverslips using a bright-field microscope. The association index was obtained by multiplying the percentage of infected macrophages by the number of amastigotes per macrophage. The 50% inhibitory concentration (IC_50_) for amastigotes was determined by a linear regression analysis. The selectivity index (SI) was calculated by dividing the CC_50_ value of THP-1 macrophage cells by the IC_50_ of amastigote forms.

### 2.15. Statistics

All experiments were performed in triplicate, in three independent experimental sets. The data were analyzed statistically by means of Student’s *t* test using GraphPad Prism 5 software (GraphPad Software, San Diego, CA, USA). *p* values of 0.05 or less were considered statistically significant.

## 3. Results and Discussion

### 3.1. Effects of Coordination Compounds on Leishmania’s Growth Rate

Initially, we evaluated the susceptibility of *L. amazonensis* and *L. chagasi* promastigotes to phen, its quinone derivative phendione, the phendione complexed with silver(I) (Ag-phendione) and copper(II) (Cu-phendione) as well as to the simple salts AgNO_3_ and CuSO_4_. To do it, promastigotes were incubated in the absence (control) or in the presence of different concentrations of each test compound and the cellular growth was compared with nontreated parasites for 72 h. For both *Leishmania* species, metal-free phendione was considerably less toxic than phen. When phendione was individually coordinated with silver or copper metals, a significant reduction in the IC_50_ values was verified (Table 1) (Appendix A). In this sense, Ag-phendione and Cu-phendione had a similar action against *L. amazonensis*, displaying IC_50_ values of 7.8 nM and 7.5 nM, respectively, while the IC_50_ values calculated for phendione and phen were 19.1 nM and 870.0 nM, respectively (Table 1). Similar results were evidenced for *L. chagasi*, in which Cu-phendione was the most effective compound (IC_50_ = 20.0 nM) followed by Ag-phendione, phendione and phen (Table 1). Additionally, promastigotes cultured either in the presence of the simple salts or DMSO (the solvent of the test compounds) had no effect on the parasite proliferation when used in the same concentration of the coordination compounds (Appendix A). 

The effects of these test compounds on other microorganisms’ growth rate were also evaluated. For instance, in a study conducted with the fungus *P. verrucosa*, an IC_50_ value of 7.0 μM was found for the phendione ligand. The addition of the transition metals silver and copper resulted in a potentiated effect of these compounds, with calculated IC_50_ values of 2.4 and 1.8 µM, respectively [11]. The effect of these metal complexes was also evaluated against the widespread and multidrug-resistant Gram-negative bacterium *P. aeruginosa*, for which the MIC_50_ values for Cu-phendione, Ag-phendione, phendione and phen were calculated as 7.43, 14.05, 31.15 and 579.28 µM, respectively [9]. Cu-phendione and Ag-phendione also affected the planktonic growth and biofilm formation of carbapenemase-producing Gram-negative bacterium *A. baumannii* [13]. These metal-phendione complexes also present a potent anti-*T. vaginalis* action when used alone or in combination with the reference drug metronidazole [12]. Recently, Lima and colleagues [17] showed that both metal-based compounds harmed the metallopeptidase gp63 activity of virulent strains of *L. braziliensis*, in addition to altering the parasite–macrophage interaction process.

### 3.2. Effects of Coordination Compounds on Leishmania’s Morphometry and Ultrastructure

The treatment of *L. amazonensis* and *L. chagasi* promastigotes with Ag-phendione and Cu-phendione, at both IC_50_ and 2 × IC_50_ concentrations, induced a significant reduction in the parasite size in comparison to untreated cells as judged by a flow cytometry analysis, while the treatment with ½ × IC_50_ did not impact this morphometrical parameter (Figure 1). In parallel, an increase in the cell granularity of *L. amazonensis* was observed after the treatment with both test compounds, while no change in this morphometric parameter was detected in *L. chagasi* under the employed experimental conditions (Figure 1). In order to analyze the morphological changes caused by the test compounds in more detail, promastigotes were treated with the IC_50_ values of Ag-phendione and Cu-phendione and examined by SEM. The results showed that *L. amazonensis* (Figure 2A) and *L. chagasi* (Figure 2F) untreated parasites showed normal ultrastructural features, including regular cell surface, elongated cell body and a long and unique flagellum. The treatment of *L. amazonensis* promastigotes with Ag-phendione and Cu-phendione led to several ultrastructural alterations, such as a rounding of the cell body, cell shrinkage and a shortening of the flagellum (Figure 2B–E). Moreover, treatment with Cu-phendione also caused a discontinuity on the cell surface of the parasites (Figure 2E). Similarly, treatment of *L. chagasi* promastigotes with the IC_50_ value of both coordination compounds caused rounding in the cell body and cell shrinkage (Figure 2G–J). In addition, the parasites presented an increase in membrane protrusion resembling surface blebs compared to the control cells (Figure 2G–J). The treatment of *L. chagasi* with Cu-phendione also promoted the complete loss or shortening of the single flagellum (Figure 2I,J).

In recent years, a series of compounds complexed to metals have been tested against *Leishmania* [22,23,24,25,26,27,28], and these new formulations may emerge as potential drugs in the treatment of leishmaniasis. A study using amino- and iminopyridyl compounds complexed with metals presented a potent effect against *Trypanosoma cruzi* and *Leishmania* [24]. Similar to the effects observed in the present work, the metal complexes induced morphological alterations such as a reduction or swelling of the cellular body and a shortening or loss of the flagellum [24]. The treatment of *L. amazonensis* with benzaldehyde thiosemicarbazone derived from limonene and complexed with copper, termed BenzCo, promoted morphological alterations in the parasite such as a reduction in the size of the parasites besides alterations in the ultrastructure [22]. Ternary nickel(II) complexes with a triazolopyrimidine derivative and different aliphatic or aromatic amines as auxiliary ligands have a potent antiproliferative effect against promastigotes and amastigotes of *L. infantum* and *L. braziliensis*, besides promoting ultrastructural alterations such as intense cytoplasmic vacuolization and alterations in the membranes forming the internal organelles [23]. The treatment of *L. amazonensis* promastigotes with biogenic silver nanoparticles (AgNp-bio), obtained by reducing silver nitrate using the nitrate reductase enzyme from *Fusarium oxysporium*, presented leishmanicidal activity at doses of 0.25 μg/mL in a time-dependent manner; furthermore, the treatment of infected peritoneal macrophages with the same concentration was also able to reduce both the number of infected macrophages and the number of intracellular amastigotes [25]. In another study, Alti and collaborators [26] showed a potent anti-*L. donovani* activity of gold–silver bimetallic nanoparticles (Au–Ag BNPs) synthesized through reduction with medicinal plant extracts and that presented IC_50_ values of 0.035 μg/mL. Similarly, gold and silver nanoparticles functionalized with 4′,7-dihydroxyflavone exhibited a promising activity against promastigote and amastigote forms of *L. donovani* [27]. Albalawi and collaborators [28] showed that copper nanoparticles (CuNPs) green-synthesized from *Capparis spinosa*’s fruit extract or combined with meglumine antimoniate (MA) suppressed the growth rate of *L. major* amastigotes in a typically dose-dependent mode. The IC_50_ values were 116.8 ± 3.05 and 21.3 ± 0.42 μg/mL for the CuNPs alone and CuNPs along with MA, respectively. Overall, these results indicate that coordination compounds can be considered potent and selective molecules against *Leishmania*. 

### 3.3. Effects of Coordination Compounds on Leishmania’s General Metabolism and Mitochondrial Activity

The toxicity of the coordination compounds on promastigote forms was tested by the colorimetric assay using resazurin dye/Alamar blue. Resazurin is a redox potential indicator dye, nontoxic for cells even during long incubation times and that has been used for the determination of cell viability [21]. In living cells, resazurin is reduced to the colorimetric resorufin dye, thereby changing its color from blue to red, while nonviable cells do not produce fluorescent signals [20]. These indicators have been used in different studies to assess the drug sensitivity of *Leishmania* promastigotes [20,29]. Herein, the results showed that both test compounds affected the global metabolism of both *L. amazonensis* and *L. chagasi* in a typically dose-dependent manner (Figure 3), displaying a statistically significant reduction when the treatment was performed at the concentrations corresponding to IC_50_ and 2 × IC_50_ values of both Ag-phendione and Cu-phendione. Parasites treated with sodium azide (40 μM) were used as a positive control of nonmetabolically active cells (Figure 3). 

In trypanosomatids, the mitochondrion corresponds to a single elongated organelle, which implies the necessity of its perfect functioning so that the protozoa can perform their vital functions [30,31]. On the other hand, the fact that mammalian cells present multiple mitochondria allows a mechanism of compensation in the presence of altered organelles. Thus, the mitochondrion of trypanosomatids acts as an attractive target for the development of new chemotherapeutics [30]. In this work, we verified the effects of the test compounds on both mitochondrial dehydrogenase activities (Figure 4) and mitochondrial membrane potential (Figure 5). 

The treatment of parasite cells with Ag-phendione and Cu-phendione induced a reduction in mitochondrial dehydrogenase activities in a concentration-dependent manner (Figure 4). In this sense, IC_50_ and 2 × IC_50_ values of Cu-phendione caused a significant reduction in the enzymatic activity of mitochondrial dehydrogenases; the highest concentration led to approximately a 50 and 75% decay for *L. amazonensis* and *L. chagasi*, respectively, when compared to untreated promastigotes (Figure 4). The same range of inhibition was detected with Ag-phendione at 2 × IC_50_ values. The mitochondrial membrane potential is a key indicator of mitochondrial functionality and viability, because it reflects the process of electron transport and oxidative phosphorylation, the driving force behind ATP production [32]. In order to analyze the effect of the test coordination compounds on the mitochondrial membrane potential, parasite cells were labeled with Rhodamine 123 and analyzed in a flow cytometer. The results indicated that the treatment of *L. amazonensis* and *L. chagasi* with IC_50_ and 2 × IC_50_ values of either Ag-phendione or Cu-phendione significantly diminished the mitochondrial membrane potential when compared to the untreated control, indicating a mitochondrial membrane depolarization (Figure 5). Contrarily, no significant difference was observed when the parasites were treated with ½ × IC_50_ values of both test compounds. Parasites incubated with FCCP, a mitochondrial protonophore, were used as a positive control of mitochondrial membrane depolarization (Figure 5). 

A similar result was found when promastigotes of *L. amazonensis* were treated with BenzCo and then labeled with Rhodamine 123 [22]. Similarly, the treatment of promastigotes of *L. amazonensis* with AgNp-bio also entailed the loss of the mitochondrial membrane potential of these parasites [25]. Gélvez and collaborators [33] verified through transmission electron microscopy that the treatment of *L. amazonensis* with nanocomposites containing silver nanoparticles (AgNPs), polyvinylpyrrolidone and MA promoted significant alterations in different organelles including myelin-like structure formation inside the mitochondrion. 

The difference in susceptibility between *L. amazonensis* and *L. chagasi* to the test compounds may be a phenotype associated with the intrinsic characteristics of the *Leishmania* species [34]. Owing to the enormous genetic variability and plasticity of the genome in the *Leishmania* genus, alterations in different metabolic pathways are frequent and are related to varied responses to the same drug as well as therapeutic failures [35]. In this sense, the differences observed in cell metabolism and mitochondrial activity may be related to the particular characteristics of each *Leishmania* species, which will help in a greater or lesser response to different environmental changes.

Previous studies have shown that compounds complexed with copper have anti-trypanosomatid activity. Mixed-chelate copper(II) complexes Casiopeins^®^ exhibited a potent anti-*T. cruzi* activity, similar to the reference drug Nifurtimox^®^ [36]. Another study showed that the dinuclear copper(II) complexes using the triazolopyrimidine derivative 7-amino-1,2,4-triazolo [1,5-a]pyrimidine (7atp) showed a potent activity against *Leishmania* spp. and *T. cruzi* [37]. Copper nanoparticles exert different mechanisms on eukaryotic cells, such as oxidative stress, coordination effects and nonhomeostasis. Nanoparticles can cross a cell membrane and, inside the cell, can directly interact with oxidative organelles such as mitochondria, inducing reactive oxygen species (ROS) production, which can promote DNA strand breaks. Furthermore, Cu^2+^ ions have the ability to promote functional protein inactivation, due to their ability to move metal ions in specific metalloproteins or form chelates with biomolecules [38,39]. A series of mixed Cu(II)-phen complexes showed anticancer properties, inducing cell death mainly through a caspase-regulated apoptotic pathway leading to ROS production, the depolarization of the mitochondrial membrane, the overexpression of many apoptotic signalers and DNA cellular damage, triggering cell death [40]. Previous studies have reported that AgNPs can affect mitochondrial functioning as well as cell nucleus integrity [41]. The majority of the energy required for proper cellular function is provided by the mitochondrion, and damage to this organelle results in decreased or inefficient energy production, which may lead to cell death [41,42]. Furthermore, AgNPs can interact with membrane mitochondrial proteins and trigger a series of biological effects including mitochondrial dysfunction, oxidative stress, the interruption of ATP synthesis, DNA damage and consequent cell death by apoptosis or necrosis [41,42].

### 3.4. Effects of Coordination Compounds on Leishmania’s Cell Cycle

Since the test compounds drastically affected the mitochondrial functionality of the parasites, in this set of experiments, we decided to verify the effect of coordination compounds on the cell cycle progression of *L. amazonensis* and *L. chagasi* through flow cytometry. The results showed that the treatment with the coordination compounds induced an increase in the percentage of cells in the sub-G_0_/G_1_ phase of the cell cycle in a concentration-dependent manner, followed by a reduction in cells in the G_0_/G_1_ phase, when compared to untreated parasites (Figure 6). The S phase did not show any significant change in the percentage of cells, while a significant reduction in the number of cells was observed in the G_2_/M phase after treatment with the 2 × IC_50_ value (Figure 6). 

Compounds synthesized from transition metals have already been associated with cell cycle disruption in *Leishmania*. The treatment of *L. donovani* with copper salisylaldoxime (CuSAL) promoted an accumulation of parasite cells in the G_1_ phase of the cell cycle, with a consequent inhibition of the entry of parasites in the S phase [43]. In another study, the treatment of *L. donovani* with green-synthesized AgNPs also promoted cell cycle arrest in the sub-G_0_/G_1_ phase and a simultaneous decrease in both S and G_2_/M phases [44]. Similarly, in the present study, we verified that the treatment of *L. amazonensis* and *L. chagasi* promastigotes also promoted cell cycle arrest, inducing a significant increase in the proportion of cells in the sub-G_0_/G_1_ phase, followed by the interruption of the G_1_, S and G_2_/M phases, with a consequent inhibition of parasite proliferation. The accumulation of cells in the sub-G_0_/G_1_ phase of the cell cycle is a strong indication of the presence of cells with fragmented DNA, characteristic of a cell death process similar to apoptosis.

### 3.5. Effects of Coordination Compounds on Leishmania’s DNA Fragmentation

The cell cycle profile shown in Figure 6 is a strong marker for the apoptotic process, especially due to the accumulation of cells in the sub-G_0_/G_1_ phase that correspond to apoptotic bodies containing DNA fragments [45]. In dying cells, an endonuclease acts by cleaving the chromosomal DNA, causing the chromatin to fragment into multiple units [45]. In this sense, the TUNEL assay was performed in order to verify whether the compounds were able to induce DNA cleavage in *Leishmania* promastigotes. The treatment of promastigotes with IC_50_ and 2 × IC_50_ values of Ag-phendione and Cu-phendione during 72 h confirmed the DNA fragmentation through the increase in the incorporation of BrdU and a consequent increase in the % FC (Figure 7). Similarly, the treatment of *L. donovani* with copper salisylaldoxime (CuSAL) for 24 h was able to promote DNA fragmentation, indicating a process of cell death by apoptosis [43]. AgNPs also induced DNA fragmentation in *L. donovani* promastigotes, as verified by the TUNEL technique; however, gel electrophoresis showed that DNA breakage was not extensive, indicating the presence of high-molecular-weight DNA fragments of 700 bp [44]. Interestingly, the treatment of *P. aeruginosa* with bactericidal concentrations of Cu-phendione induced DNA fragmentation [46]. Ag-phendione and Cu-phendione were able to bind to double-stranded DNA via hydrogen bonding, hydrophobic and electrostatic interactions. In addition, Cu-phendione was able to induce topoisomerase I mediated DNA relaxation of supercoiled plasmid DNA, in addition to inducing oxidative DNA injuries [46].

### 3.6. Effects of Coordination Compounds on Leishmania’s Phosphatidylserine Externalization

The induction of programmed cell death results in a series of biochemical events, such as an increase in the number of cells in the sub-G_0_/G_1_ phase of the cell cycle and the externalization of phosphatidylserine (PS) [41,42]. Since the treatment of *L. amazonensis* and *L. chagasi* with Ag-phendione and Cu-phendione induced the appearance of apoptotic characteristics, including a reduction in parasite size, a decrease in energy production, the arresting of cell cycle progression and DNA cleavage, we evaluated the externalization of PS in order to corroborate the hypothesis that the test coordination compounds act by inducing the apoptosis-like death pathway in *Leishmania* promastigotes. In eukaryotic cells, PS is present in the inner leaflet of the lipid bilayer and any change in this distribution causes a physiological event such as the clearance of apoptotic cells and, therefore, PS exposure has been implicated as an apoptotic marker [47]. Annexin V (AX-V) is a protein with affinity for PS, being used to evaluate cellular apoptotic processes. The translocation of PS from the inner to the outer side layer of the plasma membrane is a common phenomenon in eukaryotic cell apoptosis processes [47]. Although studies indicate that *Leishmania* does not have any detectable amounts of PS, other classes of phospholipids that bind to AX-V are present, such as phosphatidic acid, phosphatidylethanolamine; phosphatidylglycerol, phosphatidylinositol and cardiolipin [48]. Since AX-V can bind to both apoptotic and necrotic cells, due to the loss of cell membrane integrity, the simultaneous addition of PI, a nonpermeable dye that selectively binds to nucleic acids, allows one to differentiate between apoptotic cells [AX-V (+)/PI (−)] and late apoptotic/necrotic cells [AX-V (+)/PI (+)] [49]. 

In this work, we verified that the treatment of *L. amazonensis* and *L. chagasi* with the IC_50_ and 2 × IC_50_ values of Ag-phendione and Cu-phendione promoted an increase in the percentage of positive cells only for AX-V, as well as when doubly labeled with AX-V (+)/PI (+). In the presence of 2 × IC_50_ values of the coordination compounds, about 12.5% of the cells of both species of *Leishmania* were AX-V (+)/PI (−), indicating early stages of apoptotic events, and approximately 27% and 48% of *L. amazonensis* and *L. chagasi* promastigotes presented the AX-V (+)/PI (+) phenotype, respectively, characterizing events of late apoptosis and/or necrosis (Figure 8). Additionally, the treatment of *L. chagasi* with the IC_50_ value of Cu-phendione promoted a double labeling of 27% of the population. Previous studies showed that *L. amazonensis* treated with AgNp-bio also stimulated the PS exposure, as well as the loss of cell membrane integrity, as verified by double labeling with AX-V and PI [25]. Zahir and collaborators [44] showed that *L. donovani* promastigotes treated with Ag NPs for 24 h were positive for AX-V and PI or only for PI, suggesting that AgNPs induce cell death by necrosis. Similarly, an increase in the number of AX-V-positive cells was also observed after the treatment of *L. major* with green-synthesized AgNPs via ginger rhizome extract, indicating the induction of cell death via apoptosis [50].

The effect of metal-based compounds on the activation of caspase-like proteins as well as on the inhibition of antioxidant agents has also been studied as triggers of programmed cell death. The presence of caspase-like proteases, such as the CED-3/CPP32 group of proteases (caspase 3) and the ICE family of proteases (caspase 9), are well documented and play fundamental roles in the apoptotic cascade of *Leishmania* [51,52]. Compounds derived from silver and copper have already been reported in the activation of these proteases and their consequent action in triggering the process of cell death by apoptosis. The treatment of *L. donovani* with CuSAL activated the ICE family of proteases and the CED3/CPP32 group of proteases; in parallel, when in the presence of a specific inhibitor of these proteases, events derived from the caspase activation, such as DNA fragmentation, were prevented, suggesting that the activation of caspase-like proteases was involved in the CuSAL induced programmed cell death of *Leishmania* parasites [43]. Antioxidant molecules, such as trypanothione reductase (TR), participate in antioxidant reactions such as thiol–disulfide exchange, in addition to acting as electron donors in several metabolic pathways, being essential in maintaining the reduced environment and thus the survival of the parasite within the cell [44,53]. Silver ions are excellent inducers of ROS production in *Leishmania* as well as an effective TR inhibitor, leading to a process of cell death by apoptosis [44,53]. This analysis was corroborated by the treatment of *L. donovani* with green-synthesized AgNPs, which led to an increase in ROS production followed by a potent reduction in the total intracellular thiols, being accompanied by an increase in the G_0_/G_1_ phase of the cell cycle and DNA fragmentation, characteristic markers of the apoptotic death process [44].

The loss of mitochondrial membrane potential, DNA fragmentation, cell cycle arrest and PS externalization suggest that both tested coordinated compounds, Ag-phendione and Cu-phendione, induce cell death via an apoptosis-like mechanism [54]. However, new studies in order to evaluate other markers, such as cytochrome c release, endonuclease g, metacaspase and calpains [54], are needed in order to more accurately evaluate the mechanism of action of these promising compounds.

### 3.7. Effects of Coordination Compounds on THP-1 Macrophage Cells

Initially, the toxicity of Ag-phendione and Cu-phendione to THP-1 macrophages was assessed by MTT. The CC_50_ value calculated for Ag-phendione was 1870 nM and for Cu-phendione, it was 1470 nM after 24 h of treatment (Table 2). Dilutions of DMSO corresponding to the highest concentration of each compound had no effect on macrophage viability (data not shown). Based on the calculated CC_50_ values, coordination compounds showed an excellent selectivity index (SI) against the promastigote forms of both species of *Leishmania*. Ag-phendione and Cu-phendione presented an SI of 239.7 and 196.0, respectively, for *L. amazonensis*; in addition, for *L. chagasi*, the calculated values were 76.3 for Ag-phendione and 73.5 for Cu-phendione. The promising efficacy of the compounds against the promastigote forms of *Leishmania* stimulated the study of their effects on the amastigote forms of the parasites.

### 3.8. Effects of Coordination Compounds on Leishmania–Macrophage Interaction

The establishment and maintenance of the infection in the vertebrate host is a fundamental step for the development of the disease, and as such, the *Leishmania*–macrophage interaction is crucial in this process. In this work, we found that the coordination compounds were able to drastically reduce the survival of intracellular amastigotes without presenting a toxic effect to macrophages. THP-1 macrophages previously infected with *L. amazonensis* or *L. chagasi* and, subsequently, treated with the coordination compounds showed a significant reduction in the association index when compared to their respective untreated systems (Table 2). The results showed that both coordination compounds significantly suppressed the growth rate of intracellular amastigotes of *L. amazonensis* and *L. chagasi*. The obtained IC_50_ values for *L. amazonensis* were 43 nM for Ag-phendione and 35 nM for Cu-phendione, while *L. chagasi* showed IC_50_ values of 88 and 51 nM for Ag-phendione and Cu-phendione, respectively. Under these employed conditions, both Ag-phendione and Cu-phendione showed a higher toxicity to *Leishmania* amastigotes compared to THP-1 macrophage cells, resulting in excellent SI values (Table 2). Albalawi and collaborators [28] verified that green-synthesized CuNPs alone or combined with MA presented SI values of 11.34 and 18.60, respectively, against intramacrophage amastigote forms of *L. major* [28]. Green-synthesized AgNPs showed a low toxicity against J774A.1 macrophages with a CC_50_ of 115.5 µg/mL, while for intracellular amastigotes of *L donovani*, an IC_50_ value equivalent to 3.89 µg/mL was detected [44]. In another study, AgNp-bio did not present cytotoxicity on peritoneal macrophages at concentrations of 0.125, 0.25 and 0.50 μg/mL, indicating a good tolerability against macrophages infected with *L. amazonensis* [25]. Toxicity studies using RAW 264.7 macrophages revealed that a treatment with CuSAL at 30 μM for 8 h and 12 h reduced the number of intracellular amastigotes by 97% and 100%, respectively, without presenting any toxicity to the host cell [43].

Previously, our group found that the pretreatment of *L. braziliensis* promastigotes for 1 h with Ag-phendione and Cu-phendione induced a decrease in the association index with macrophages by 51.4% and 44.4%, respectively [17]. Vargas Rigo and collaborators [12] evaluated that Ag-phendione and Cu-phendione showed excellent selectivity for *T. vaginalis* compared to HMVII vaginal epithelial cells, erythrocytes and nontumor cell line 3T3-C1 and that Cu-phendione was the compound best tolerated in vitro by host cells, indicating the selectivity for the parasite and the safety of the complex. Previous studies indicated that Cu-phendione and Ag-phendione promoted a protective action on lung epithelial cells (A549) previously incubated with a *P. aeruginosa* supernatant, which is rich in elastase B (LasB), an important virulence factor of this bacterium. In addition, Cu-phendione was able to reduce the toxic effects of LasB in the *G. mellonella* model [55]. McCann and coworkers [6] had already verified that these compounds presented an excellent tolerability against different cell lineages. Studies conducted in vivo using the larvae of the insect *G. mellonella* indicated that treatment with up to 33.3 mg/kg of these metal-based compounds ensured 100% of survival of the larvae. In addition, studies using Swiss mice indicated that treatment with a concentration of up to 45 mg/kg/day ensured the survival of mice after 7 days of treatment. Moreover, blood samples taken from Swiss mice revealed that treatment with Ag-phendione and Cu-phendione did not affect the levels of two hepatic enzymes, aspartate aminotransferase and alanine aminotransferase, reinforcing the selective action of both coordination compounds against *Leishmania* parasites.

The mechanism of action induced by these test coordination compounds in trypanosomatids is still unknown. In this work, we proposed the mitochondrion as one of the major targets, as well as an arrest in the cell cycle, PS exposure and DNA fragmentation, followed by a cell death via apoptosis. Previous studies by our group [17] showed that these compounds also acted on the gp63 metalloprotease (an important virulence factor) of *Leishmania* promastigotes, an essential molecule in the parasite internalization process. The reduction in the expression of these molecules promoted by the treatment of promastigotes with Ag-phendione and Cu-phendione may be directly related to the reduction in the number of intramacrophagic amastigotes. Furthermore, the loss of mitochondrial viability as well as the expression of apoptotic/necrotic characteristics after treatment of promastigotes may be acting together to reduce the infection. Therefore, further studies are needed to determine the mechanisms responsible for the reduction of intracellular amastigotes.

## 4. Conclusions

Drugs currently available for the treatment of leishmaniasis have serious side effects and are often ineffective; in addition, an increasing number of cases of resistance has been reported [3,4]. The beneficial picture of phendione-based compounds [6,7,8] together with the need to develop new chemotherapeutic options for the prevention and/or treatment of infections caused by *Leishmania* led us to broaden our studies in an attempt to decipher the mechanisms of action of Ag-phendione and Cu-phendione against *L. amazonensis* and *L. chagasi*, which are etiological agents of cutaneous and visceral leishmaniasis, respectively.

In conclusion, Ag-phendione and Cu-phendione show a high potency in inhibiting the growth of promastigote forms of *L. amazonensis* and *L. chagasi*, in addition to altering the morphology of these parasites. The findings also indicated that the antileishmanial mechanisms of Ag-phendione and Cu-phendione point to an apoptosis-like cell death process due to the changes in mitochondrial viability/membrane potential, cell cycle arrest in sub-G_0_/G_1_ phase, DNA damage and annexin-V staining. Furthermore, the coordination compounds also showed a potent effect against the amastigote forms of both *Leishmania* species, with reduced toxicity to the host cell (presenting excellent selectivity indexes). Previous studies from our group have shown that these metal-based drugs have excellent tolerability to different cell lines as well as in in vivo animal models [6], confirming that they stand out as a new therapeutic option for the treatment of cutaneous and visceral leishmaniasis.

## Figures and Tables

**Figure 1 pathogens-12-00070-f001:**
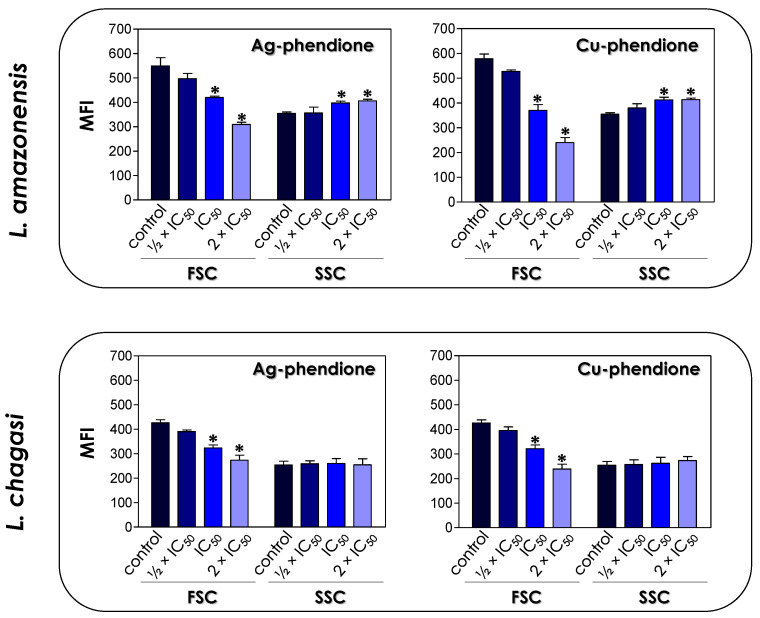
Effects of coordination compounds on the cell size (FSC) and granularity (SSC) of *L. amazonensis* and *L. chagasi* promastigote forms. Parasites were treated or not (control) with the ½ × IC_50_, IC_50_ and 2 × IC_50_ values of Ag-phendione and Cu-phendione for 72 h and, subsequently, analyzed in a flow cytometer. FSC and SSC values were expressed as the mean of the fluorescence intensity (MFI). The data presented in the graph are representative of the analysis of 10,000 parasite cells in three experiments performed in triplicate and the results were considered significant when *p* < 0.05 (*).

**Figure 2 pathogens-12-00070-f002:**
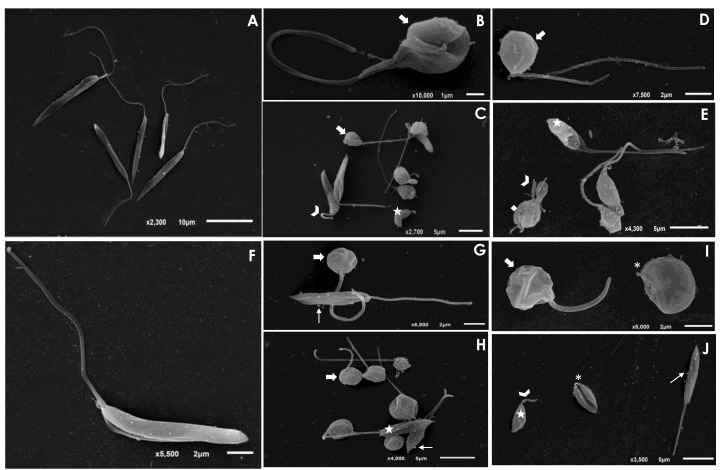
Scanning electron microscopy (SEM) analysis of *L. amazonensis* and *L. chagasi* promastigotes after treatment with the IC_50_ value of Ag-phendione and Cu-phendione. Control cells of *L. amazonensis* (**A**) and *L. chagasi* (**F**) presented an elongated cell body and a long flagellum. The treatment of *L. amazonensis* with the IC_50_ value of Ag-phendione (**B**,**C**) and Cu-phendione (**D**,**E**) showed parasites displaying rounding in cell body (arrows), cell shrinkage (stars) and shortening of flagellum (arrowheads). In addition, treatment with Cu-phendione led to a discontinuity on the cell surface (diamond). *L. chagasi* treated with Ag-phendione (**G**,**H**) and Cu-phendione (**I**,**J**) also promoted rounding in cell body (thick arrows) and cell shrinkage (stars), besides the formation of blebs in the membrane (thin arrows). *L. chagasi* treated with Cu-phendione presented shortening (arrowhead) or loss (asterisks) of the flagellum.

**Figure 3 pathogens-12-00070-f003:**
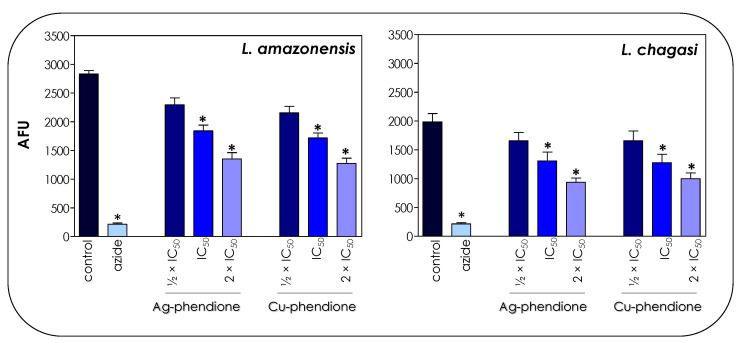
Effects of coordination compounds on the general metabolism of *L. amazonensis* and *L. chagasi* promastigote forms. Parasites were incubated in the absence (control) and in the presence of Ag-phendione and Cu-phendione (½ × IC_50_, IC_50_ and 2 × IC_50_ values) for 72 h and, subsequently, incubated with resazurin. The reduction of resazurin was expressed as arbitrary fluorescence units (AFU). Parasites treated with sodium azide (40 µM) were used as a positive control of nonviable cells. The results were considered statistically significant when *p* < 0.05 (*).

**Figure 4 pathogens-12-00070-f004:**
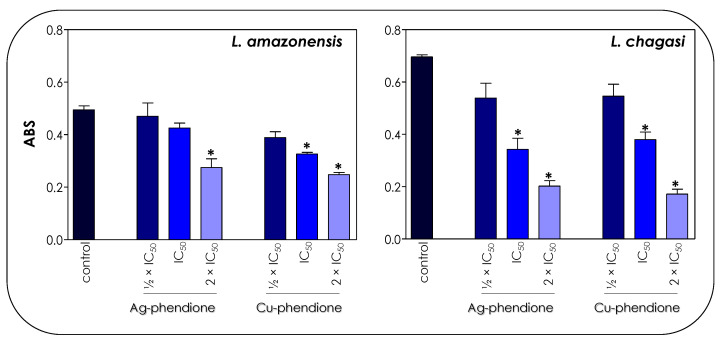
Effects of coordination compounds on the mitochondrial metabolism of *L. amazonensis* and *L. chagasi* promastigotes. Parasites were incubated in the absence (control) and in the presence of the ½ × IC_50_, IC_50_ and 2 × IC_50_ values of Ag-phendione and Cu-phendione for 72 h and, subsequently, incubated with MTT for 4 h at 37 °C. Mitochondrial viability was determined by absorbance (ABS) quantification at 570 nm and the results were considered statistically significant when *p* < 0.05 (*).

**Figure 5 pathogens-12-00070-f005:**
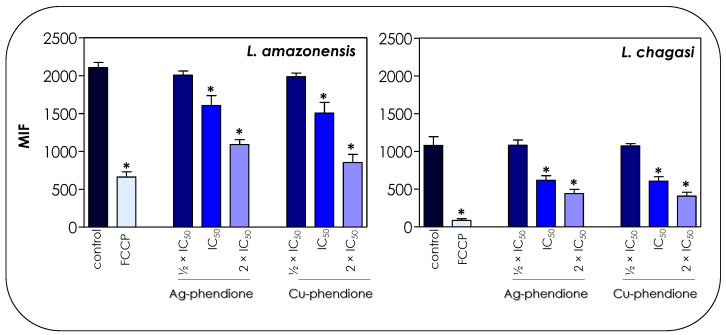
Effects of coordination compounds on the mitochondrial membrane potential of *L. amazonensis* and *L. chagasi* promastigotes. Parasites were pretreated or not (control) for 72 h with the ½ × IC_50_, IC_50_ and 2 × IC_50_ values of Ag-phendione and Cu-phendione and incubated with Rhodamine 123 (10 μg/mL) for 20 min before analysis on a flow cytometer. FCCP was used as a membrane depolarization control. The data were expressed as the mean of the fluorescence intensity (MFI) and are representative of the analysis of 10,000 cells in experiments performed in triplicate. The results were considered statistically significant when *p* < 0.05 (*).

**Figure 6 pathogens-12-00070-f006:**
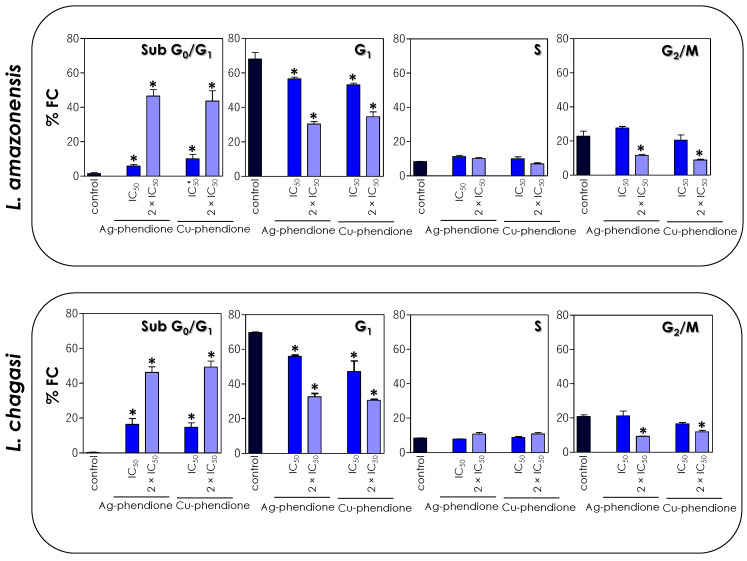
Effects of coordination compounds on the cell cycle arrest of *L. amazonensis* and *L. chagasi* promastigotes. Parasites were incubated in the absence (control) and in the presence of the IC_50_ and 2 × IC_50_ values of Ag-phendione and Cu-phendione for 72 h and, subsequently, stained with PI and analyzed by flow cytometry. The graphs show the percentage of fluorescent cells (% FC) in the different phases of the cell cycle (sub-G_0_/G_1_, G_1_, S and G_2_/M). The data are representative of the analysis of 10,000 cells in experiments performed in triplicate, and the results were considered significant when *p* < 0.05 (*).

**Figure 7 pathogens-12-00070-f007:**
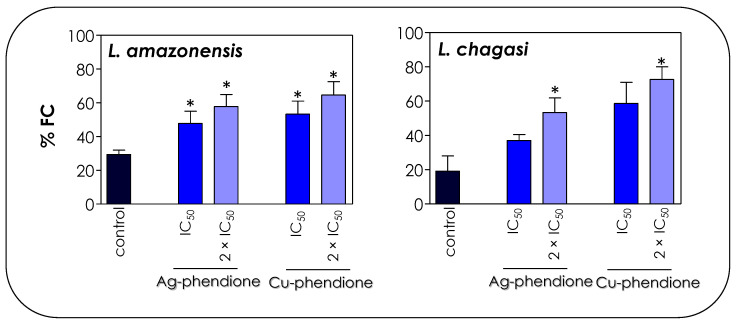
Effects of coordination compounds on the DNA fragmentation of *L. amazonensis* and *L. chagasi* promastigotes. Parasites were incubated in the absence (control) and in the presence of the IC_50_ and 2 × IC_50_ values of Ag-phendione and Cu-phendione for 72 h and, subsequently, stained with dUTP-FITC in the presence of terminal deoxynucleotidyl transferase and RNase enzyme, followed by flow cytometry analysis. The graphs show the percentage of fluorescent cells (% FC). The data are representative of the analysis of 10,000 cells in experiments performed in triplicate, and the results were considered significant when *p* < 0.05 (*).

**Figure 8 pathogens-12-00070-f008:**
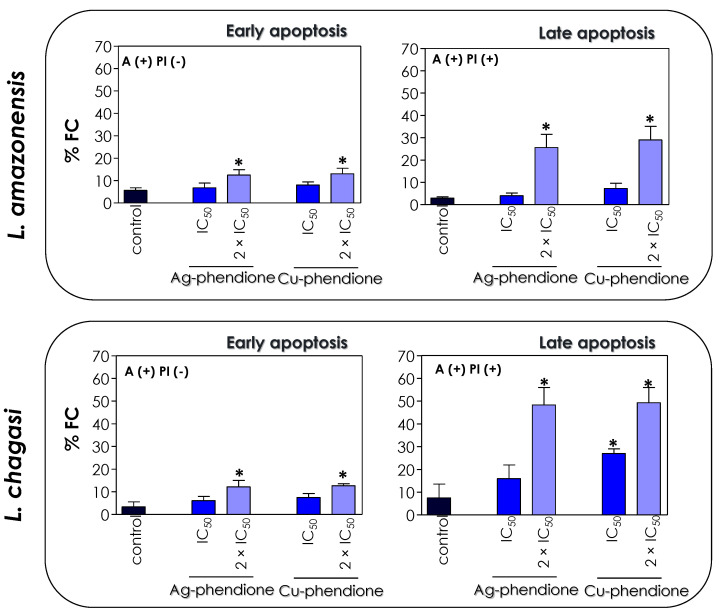
Effects of coordination compounds on the expression of apoptosis-associated markers in *L. amazonensis* and *L. chagasi* promastigotes. Parasites were incubated in the absence (control) and in the presence of the IC_50_ and 2 × IC_50_ values of Ag-phendione and Cu-phendione for 72 h and subsequently co-stained with PI and annexin V–Alexa Fluor 488 and analyzed by flow cytometry. The graphs show the percentage of fluorescent cells (% FC) for annexin V [A(+) PI(−)] and for annexin V and PI [A(+) PI (+)]. The data are representative of the analysis of 10,000 cells in experiments performed in triplicate, and the results were considered significant when *p* < 0.05 (*).

**Table 1 pathogens-12-00070-t001:** Anti-*Leishmania* effects of Ag-phendione, Cu-phendione, phendione and phen.

	IC_50_ Values (nM)	
Compounds	*L. amazonensis*	*L. chagasi*
Cu-phendione	7.5	20.0
Ag-phendione	7.8	24.5
Phendione	19.1	30.0
Phen	870.0	1120.0

Phen, phendione, Ag-phendione and Cu-phendione represent 1,10-phenanthroline, 1,10-phenanthroline-5,6-dione, [Ag(phendione)_2_]ClO_4_ and [Cu(phendione)_3_](ClO_4_)_2_·4H_2_O, respectively. IC_50_, 50% inhibitory concentration.

**Table 2 pathogens-12-00070-t002:** Cytotoxicity effects of Ag-phendione and Cu-phendione against THP-1 macrophages and anti-amastigote activity.

	THP-1 Cells	*L. amazonensis*		*L. chagasi*	
Compounds	CC_50_ (nM)	IC_50_ (nM)	SI	IC_50_ (nM)	SI
Cu-phendione	1470	35	42	51	28.8
Ag-phendione	1870	43	43.4	88	21.2

Ag-phendione and Cu-phendione represent [Ag(phendione)_2_]ClO_4_ and [Cu(phendione)_3_](ClO_4_)_2_·4H_2_O, respectively. CC_50_, 50% cytotoxic concentration. IC_50_, 50% inhibitory concentration. SI, selectivity index.

## Data Availability

Not applicable.

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
