# Peer review of "The Anti-*Leishmania amazonensis* and Anti-*Leishmania chagasi* Action of Copper(II) and Silver(I) 1,10-Phenanthroline-5,6-dione Coordination Compounds"

_pathogens, 2023, doi:10.3390/pathogens12010070_

Round 1
Reviewer 1 Report
The manuscript authored by Oliveira et al. presents interesting and important results about the anti-leishmania activity of two metal coordinate compounds. The authors show that the IC50 and 2x IC50 of the compounds have drastic effects on promastigotes' cell morphology, cell membrane, cell cycle, cell growth and proliferation, cell survival (they induced apoptosis and or necrosis) and induce mitochondria alterations (metabolism and membrane potential), in two species of Leishmania, L. amazonensis and L. infantum. The compounds could also induce a decrease in intracellular amastigotes proliferation of the two parasite species, without causing any effect on the macrophage lineage, demonstrating selective index values. But, some of the results need further discussion, such as the results presented in figures 3, 4, and 5. In these specific cases, is clear that there are intra- and inter-species differences concerning the activity of the compounds. Here is worth reminding that the effects are against Leishmania species that cause different clinical forms of the disease. The title of the manuscript also deserves a revision. I propose - “The anti-Leishmania action of two metal-coordinate compounds containing 1,10-2 phenanthroline-5,6-dione”.
Major
.Materials and methods
- - How many in vitro passages do L. amazonensis and L. infantum promastigotes had when used for the in vitro experiments?
- - How was chosen the starting concentration of the metal-coordinated and metal-free compounds?
- Protocol assay- Line 222- L. amazonensis promastigotes usually enter stationary phase on day 4-5 and L. chagasi on day 4. How do they know that on day 3 of culture both species have a good percentage of metacyclics. Did they use any kind of metacyclics selection? It is recommended to show the growth curves for both species as supplementary material.
.Results
Line 303- WT images also show "blebs" in the cell membrane- what do blebs mean???? Bacteria contamination in the cultures, in the preparation solutions?
Item 3.3- Include a discussion about the effects of Cu-coordinated drugs against Leishmania and other tryps. If there are any results published you should mention them!
It is recommended that all flow cytometer results (histograms and graphs generated by Flow Jo) be presented as Supplementary Material
Results presented in Fig. 7 and Table 2 should be complemented with cell images
Minor
Line 91- include a reference
Line 108- No antibiotics or antimycotics were added to the culture medium.(?)
Line 136- GraphPad version?
Line 157 and 168- In both protocols parasites were used free of culture medium and washed and incubated in which buffer?
Lines 244, 284 -italicize “Leishmania”
Author Response
Authors’ Comments to Reviewer 1
Reviewer 1: The manuscript authored by Oliveira et al. presents interesting and important results about the anti-leishmania activity of two metal coordinate compounds. The authors show that the IC50 and 2x IC50 of the compounds have drastic effects on promastigotes' cell morphology, cell membrane, cell cycle, cell growth and proliferation, cell survival (they induced apoptosis and or necrosis) and induce mitochondria alterations (metabolism and membrane potential), in two species of Leishmania, L. amazonensis and L. infantum. The compounds could also induce a decrease in intracellular amastigotes proliferation of the two parasite species, without causing any effect on the macrophage lineage, demonstrating selective index values. But, some of the results need further discussion, such as the results presented in figures 3, 4, and 5. In these specific cases, is clear that there are intra- and inter-species differences concerning the activity of the compounds. Here is worth reminding that the effects are against Leishmania species that cause different clinical forms of the disease. The title of the manuscript also deserves a revision. I propose - “The anti-Leishmania action of two metal-coordinate compounds containing 1,10-2 phenanthroline-5,6-dione”.
Authors: Firstly, the authors are grateful for the reviewer’s kind comments and valuable suggestions on the manuscript. The title has been changed, as suggested. In addition, a paragraph describing the differences between Leishmania species was added (pages 11-12).
Major
Materials and methods
Reviewer 1: How many in vitro passages do L. amazonensis and L. infantum promastigotes had when used for the in vitro experiments?
Authors: For the experiments carried out in this work, we did not use recently isolated parasites; therefore, the number of in vitro passages was not counted. The parasites, however, are constantly used to infect macrophage cells in order to maintain their infectivity ability to mammalian cells.
Reviewer 1: How was chosen the starting concentration of the metal-coordinated and metal-free compounds?
Authors: We used an initial concentration of 10 µM for Ag-phendione, Cu-phendione and phendione and 500 µM for 1,10-phenanthroline. Since the compounds are solubilized in DMSO, the selected concentrations were based on the volume used to set up the experiments, in order to guarantee the viability of the promastigotes.
Reviewer 1: Protocol assay- Line 222- L. amazonensis promastigotes usually enter stationary phase on day 4-5 and L. chagasi on day 4. How do they know that on day 3 of culture both species have a good percentage of metacyclics. Did they use any kind of metacyclics selection? It is recommended to show the growth curves for both species as supplementary material.
Authors: Under the conditions used during our experiments, we were unable to reach the stationary growth phase of L. amazonensis and L. chagasi, and it was not possible to quantify the percentage of metacyclics. Growth curves of L. amazonensis and L. chagasi have been added as Figure S1.
Results
Reviewer 1: Line 303- WT images also show "blebs" in the cell membrane- what do blebs mean???? Bacteria contamination in the cultures, in the preparation solutions?
Authors: Blebs are protrusions formed by budding or shedding from the plasma membrane and that vary in size and density, which is a very common physiological process in trypanosomatids. In fact, the control parasites also present blebs in the membrane, while the treatment with the compounds indicates an apparent increase in their production. This information has been added to the text (page 7).
Reviewer 1: Item 3.3- Include a discussion about the effects of Cu-coordinated drugs against Leishmania and other tryps. If there are any results published you should mention them!
Authors: Information has been added to the text (page12).
Reviewer 1: It is recommended that all flow cytometer results (histograms and graphs generated by Flow Jo) be presented as Supplementary Material
Authors: These results are attached to the reviewers, however the authors do not find it necessary to include in the publication as supplementary figures, since the data are repetitive.
Reviewer 1: Results presented in Fig. 7 and Table 2 should be complemented with cell images
Authors: We agree that the images would improve the analysis, but we are unable to carry out the experiments at this moment.
Minor
Reviewer 1: Line 91- include a reference
Authors: References have been added in the text.
Reviewer 1: Line 108- No antibiotics or antimycotics were added to the culture medium.(?)
Authors: No antibiotics or antimycotics were added to the culture medium. The medium was supplemented with fetal bovine serum only.
Reviewer 1: Line 136- GraphPad version?
Authors: GraphPad Prism 5.
Reviewer 1: Line 157 and 168- In both protocols parasites were used free of culture medium and washed and incubated in which buffer?
Authors: The reactions were carried out in the cell culture medium, in order to guarantee the ideal conditions for cell viability during the incubation period (4 h for both experiments).
Reviewer 1: Lines 244, 284 -italicize “Leishmania”
Authors: As suggested, it was italicized.

Reviewer 2 Report
well written article and relevant
Author Response
Authors’ Comments to the Reviewer 2
Reviewer 2: Well written article and relevant
Authors: The authors are grateful for the reviewer’s kind comments on the manuscript.
Reviewer 3 Report
Globally, I consider that the overall aim of the manuscript is interesting because there is no vaccine available for human leishmaniasis and the control of this infection is based on chemotherapy which showed some limitations. Thus, the search for new molecules becomes a necessity.
Although it is a nice manuscript, several major points must be revised and several questions still need some answers. Hereafter the authors will find my comments.
- Title: Why use remarkable? I suggest to delete it. Furthermore, the species responsible for cutaneous and visceral leishmaniases are numerous. I therefore suggest specifying in the title the two Leishmania species used.
- Line 53, page 2: The word (glucantime) and (pentostan) must be marked with the R for registered trademark ®.
- Line 166, page 4: the authors used 10 μM of sodium azide as a positive control in the viability test. However, in line 175, page 4, the authors used another concentration (40μM). Could you explain this?
- In the Material & Methods section, the authors should specify the origin of the reagents and materials used. Indeed, sometimes the country is specified and sometimes not!
- The word "Leishmania" should be italicized. It must be corrected throughout the manuscript.
- In paragraph 2.14, § (Materials & Methods), the authors used the ratio of 10 parasite per macrophage and 24 hours for internalization. A shorter duration is sufficient (2 to 4 hours) for infecting macrophages. Have you looked at the state of your cells after 24 hours of contact with the parasites? Why use this long contact time with a very high ratio?
- In the § (results and discussion), the authors used salts and DMSO as a control to analyze the promastigotes viability. However, the results obtained are not presented (data not shown). It would have been more interesting to presente the % viability of the promastigotes in addition to the IC50 presented in table 1.
- Why the authors did not used the pentavalent antimonials as a referent control to better appreciate the effectiveness of these molecules on promastigotes growth?
- Before saying that it is apoptosis death, it is necessary to test whether these molecules induce the activation of caspase 3. It will also be interesting to make an agarose gel to assess DNA fragmentation after treatment.
- It would be important for the authors to discuss a little about the mechanism induced by coordination compounds on Leishmania. The potential mechanisms of these compounds to reduce the intramacrophagic amastigotes? What is the added value of silver and copper?
Author Response
Authors’ Comments to the Reviewer 3
Reviewer 3: Globally, I consider that the overall aim of the manuscript is interesting because there is no vaccine available for human leishmaniasis and the control of this infection is based on chemotherapy which showed some limitations. Thus, the search for new molecules becomes a necessity.
Although it is a nice manuscript, several major points must be revised and several questions still need some answers. Hereafter the authors will find my comments.
Authors: The authors are grateful for the reviewer’s kind comments and valuable suggestions on the manuscript.
Reviewer 3: Title: Why use remarkable? I suggest to delete it. Furthermore, the species responsible for cutaneous and visceral leishmaniases are numerous. I therefore suggest specifying in the title the two Leishmania species used.
Authors: The title has been modified.
Reviewer 3: Line 53, page 2: The word (glucantime) and (pentostan) must be marked with the R for registered trademark ®.
Authors: The registered trademark was added to the words.
Reviewer 3: Line 166, page 4: the authors used 10 μM of sodium azide as a positive control in the viability test. However, in line 175, page 4, the authors used another concentration (40 μM). Could you explain this?
Authors: A typing error has occurred. A volume of 10 µl of sodium azide was added in order to obtain a final concentration of 40 µM in both experiments. The value has been corrected in the text (page 4).
Reviewer 3: In the Material & Methods section, the authors should specify the origin of the reagents and materials used. Indeed, sometimes the country is specified and sometimes not!
Authors: The origin of the reagents was added to the text.
Reviewer 3: The word "Leishmania" should be italicized. It must be corrected throughout the manuscript.
Authors: As suggested, it was italicized.
Reviewer 3: In paragraph 2.14, § (Materials & Methods), the authors used the ratio of 10 parasite per macrophage and 24 hours for internalization. A shorter duration is sufficient (2 to 4 hours) for infecting macrophages. Have you looked at the state of your cells after 24 hours of contact with the parasites? Why use this long contact time with a very high ratio?
Authors: We tested a shorter time interval for macrophage infection; however, since we did not use freshly isolated parasites, the infectivity rate was low. Therefore, the 24-h period was necessary to obtain a good infection rate without altering the morphology and viability of THP-1 cells.
Reviewer 3: In the § (results and discussion), the authors used salts and DMSO as a control to analyze the promastigotes viability. However, the results obtained are not presented (data not shown). It would have been more interesting to present the % viability of the promastigotes in addition to the IC50 presented in table 1.
Authors: As included in the text (page 6), the presence of either the simple salts or DMSO had no effect on the parasite proliferation when used in the same concentration of the coordination compounds. To reinforce this, the reviewer can check Figure S1, included in the revised version of the manuscript. The IC50 value was the parameter used throughout the work to compare the results obtained.
Reviewer 3: Why the authors did not used the pentavalent antimonials as a referent control to better appreciate the effectiveness of these molecules on promastigotes growth?
Authors: At the present time, the objective of this study was to verify whether Ag-phendione and Cu-phendione had an effect on the viability of promastigotes and amastigotes of L. amazonensis and L. chagasi. As a future perspective, we intend to evaluate the effect of these compounds during in vivo infection; in this case, we will use a reference drug, such as antimonials, in order to establish a comparison regarding efficacy, tolerability and toxicity.
Reviewer 3: Before saying that it is apoptosis death, it is necessary to test whether these molecules induce the activation of caspase 3. It will also be interesting to make an agarose gel to assess DNA fragmentation after treatment.
Authors: In this work we are proposing a mechanism of cell death via apoptosis, however the authors agree that more experiments are needed to confirm the mechanism of cell death promoted by the compounds. Information has been added to the text (page 16). We also agree that the agarose gel would improve the work, but we were unable to carry out the experiment at the moment.
Reviewer 3: It would be important for the authors to discuss a little about the mechanism induced by coordination compounds on Leishmania. The potential mechanisms of these compounds to reduce the intramacrophagic amastigotes? What is the added value of silver and copper?
Authors: Information has been added to the text (page 17).
Round 2
Reviewer 3 Report
The authors have corrected errors and answered questions raised.